# Chronic Kidney Disease, Gut Dysbiosis, and Constipation: A Burdensome Triplet

**DOI:** 10.3390/microorganisms8121862

**Published:** 2020-11-25

**Authors:** Ryota Ikee, Naomi Sasaki, Takuji Yasuda, Sawako Fukazawa

**Affiliations:** 1Sapporo Nephrology Satellite Clinic, 9-2-15, Hassamu 6-jo, Nishi-ku, Sapporo 063-0826, Japan; takuji.snc@email.plala.or.jp; 2Sapporo Nephrology Clinic, 20-2-12, Nishimachikita, Nishi-ku, Sapporo 063-0061, Japan; naomi.s@email.plala.or.jp (N.S.); sawako.snc@email.plala.or.jp (S.F.)

**Keywords:** gut microbiota, dysbiosis, chronic kidney disease, constipation, intestinal motility, gut-derived uremic toxins, inflammation, short-chain fatty acids, butyrate, gut epithelial integrity

## Abstract

Gut dysbiosis has been implicated in the progression of chronic kidney disease (CKD). Alterations in the gut environment induced by uremic toxins, the dietary restriction of fiber-rich foods, and multiple drugs may be involved in CKD-related gut dysbiosis. CKD-related gut dysbiosis is considered to be characterized by the expansion of bacterial species producing precursors of harmful uremic toxins, such as indoxyl sulfate and p-cresyl sulfate, and the contraction of species generating beneficial short-chain fatty acids, such as butyrate. Gut-derived uremic toxins cause oxidative stress and pro-inflammatory responses, whereas butyrate exerts anti-inflammatory effects and contributes to gut epithelial integrity. Gut dysbiosis is associated with the disruption of the gut epithelial barrier, which leads to the translocation of endotoxins. Research on CKD-related gut dysbiosis has mainly focused on chronic inflammation and consequent cardiovascular and renal damage. The pathogenic relationship between CKD-related gut dysbiosis and constipation has not yet been investigated in detail. Constipation is highly prevalent in CKD and affects the quality of life of these patients. Under the pathophysiological state of gut dysbiosis, altered bacterial fermentation products may play a prominent role in intestinal dysmotility. In this review, we outline the factors contributing to constipation, such as the gut microbiota and bacterial fermentation; introduce recent findings on the pathogenic link between CKD-related gut dysbiosis and constipation; and discuss potential interventions. This pathogenic link needs to be elucidated in more detail and may contribute to the development of novel treatment options not only for constipation, but also cardiovascular disease in CKD.

## 1. Introduction

Chronic kidney disease (CKD) is a condition with a gradual loss of kidney function over time. Accumulated body fluids, wastes, and toxins induce anasarca, nausea, appetite loss, itchiness, fatigue, pulmonary edema, arrhythmia, and consciousness disturbance. Patients with end-stage kidney disease (ESKD) need to be treated with renal replacement therapy, such as hemodialysis (HD), peritoneal dialysis (PD), and kidney transplantation. CKD is a worldwide public health problem, and its leading cause is diabetes mellitus (DM).

Constipation is a common complication in CKD patients. In comparisons with age- and sex-matched community subjects, Cano et al. reported a higher prevalence of constipation in dialysis patients (31.1% vs. 10.1%, *p* < 0.001) [1]. The prevalence of constipation previously reported in dialysis patients ranged between 14.2% and 71.7% depending on its definition/criteria [2]. In a review by Murtagh et al., constipation was identified as the third most common symptom among the various symptoms exhibited by ESKD patients [3]. In this population, cardiovascular disease (CVD) and severe infection have a negative impact on patient outcomes [4,5,6], whereas constipation does not. Therefore, nephrologists and dialysis clinicians have not paid much attention to constipation. However, recent population-based studies reported a relationship between constipation and increased risks of colorectal cancer [7], CVD [8,9], CKD [10], and mortality [11].

Under physiological conditions, the colon receives approximately 1.5 L of liquid effluent daily from the small intestine, with 200–400 mL being excreted in stools [12]. The removal of water from fecal slurry is time-dependent and actively regulated. Sodium is reabsorbed from luminal contents through several active transport channels, with water following passively in response to osmotic gradients. On the other hand, colonic secretion is mediated through chloride channels, which are generally quiescent, leading to a net absorption of water and electrolytes [12]. Therefore, stools that remain in the colon longer become drier and harder. CKD-related constipation is primarily characterized by decreased intestinal motility (slow transit constipation), which is reflected by hard stools classified as stool forms 1–2 of the Bristol Stool Form Scale (BSFS) [13]. Wu et al. reported that the total colonic transit time was significantly longer in HD patients than in healthy controls (43.0 ± 22.2 vs. 24.3 ± 11.9 h, *p* < 0.05) [14]. A large number of factors are considered to be vital for normal intestinal motility, such as the nervous system (enteric, autonomic, and central), immune system, endocrine system, bile acid metabolism, mucus secretion, the products of colonic fermentation, and the gut microbiota [15]. In our recent review, we highlighted CKD-related constipation and discussed its specific pathogenesis [2]; decreased physical activity, various comorbidities (DM, cerebrovascular disease, and autonomic neuropathy), a decreased intake of fiber-rich vegetables and fruits, oxidative stress, hyperhomocysteinemia, and multiple drugs (potassium-binding resins, phosphate binders, and antihypertensive agents) may be involved. However, it currently remains unclear whether these factors fully explain the high prevalence of constipation in CKD patients. Gut dysbiosis, which refers to alterations in the composition and function of the gut microbiota, is increasingly being recognized along with the concept of the gut–kidney axis. Recent studies suggested pathogenic mechanisms to explain the relationship between constipation and gut dysbiosis in CKD. It is time to reconsider CKD-related constipation and discover potential clues for its successful treatment.

## 2. Gut Microbiota

The human gastrointestinal tract harbors an abundant and diverse community of trillions of microbes that mainly consists of bacteria. Examples of bacterial taxonomy in the gut are shown in Figure 1. The gut microbiota has a symbiotic relationship with the host with a broad range of benefits, such as the digestion of complex dietary macronutrients, the production of nutrients and vitamins, protection against pathogens, and the regulation of host immunity. The dominant bacterial phyla are Firmicutes, Bacteroidetes, Actinobacteria, Fusobacteria, Proteobacteria, and Verrucomicrobia, with the two phyla Firmicutes and Bacteroidetes representing 90% of the gut microbiota [16]. The phylum Firmicutes comprises more than 200 genera, such as Lactobacillus, Clostridium, Enterococcus, and Ruminococcus. Two families of this phylum, Lachnospiraceae and Ruminococcaceae, are abundant in the colon, accounting for 50–70% of bacteria in the fecal samples of healthy adults [17]. The phylum Bacteroidetes is dominated by the genera Bacteroides and Prevotella. The phylum Actinobacteria is less abundant and mainly represented by the genus Bifidobacterium. The richness and diversity of the gut microbiota are shaped in early life and characterize a healthy composition of the gut microbiota. This optimal healthy composition differs for each individual and this difference is attributable to age, ethnicity, lifestyle, and dietary habits [16]. Among these factors, dietary habits strongly impact the bacterial composition. A fiber-rich diet is associated with Prevotella-dominated microbiota, and a protein-rich diet with Bacteroides-dominated microbiota [18]. Fermented foods, alcohol, and artificial sweeteners are also suggested to influence the bacterial composition [19,20].

Gut dysbiosis is associated with a number of disorders, including inflammatory bowel disease, colorectal cancer, obesity, DM, CVD, Alzheimer’s disease, Parkinson’s disease, autism, psychological stress, and CKD [16,20]. In non-CKD subjects, constipation is associated with increases in Bacteroidetes and decreases in Bifidobacterium and Lactobacillus [15].

## 3. Fermentation Process in the Colon

Colonic bacteria obtain energy by fermenting dietary components that escape digestion in the small intestine. Two main types of bacterial fermentation are saccharolytic (carbohydrates) and proteolytic (proteins). Carbohydrate fermentation mainly occurs in the proximal part of the colon, and protein fermentation in the distal part [21].

### 3.1. Carbohydrate Fermentation and Short-Chain Fatty Acids

It is estimated that 20–60 g of dietary carbohydrates reaches the colon daily, and their main categories are resistant starch, plant cell wall polysaccharides, and indigestible oligosaccharides [17]. Complex carbohydrates are converted to monosaccharides and oligosaccharides and then fermented to hydrogen, carbon dioxide, methane, ethanol, and short-chain fatty acids (SCFAs). An increased intake of complex carbohydrates may enhance fermentative activity, resulting in increased acid production and a decreased luminal pH in the gut. An in vitro study reported that within the community at a mildly acidic pH, the Bacteroides population decreased, whereas that of the butyrate-producing Firmicutes increased [22].

SCFAs are organic weak acids characterized by containing fewer than six carbons in a straight- or branched-chain formation. The main sources of SCFAs are complex carbohydrates. However, valine, leucine, and isoleucine may be converted into branched-chain SCFAs (BCFAs). The ratio of SCFA concentrations in the colonic lumen is approximately 60% acetate, 25% propionate, and 15% butyrate [23]. SCFAs are involved in various physiological processes in the host, such as glucose homeostasis, appetite regulation, lipid metabolism, and immune regulation [24]. In addition, colonocytes derive a large proportion of energy from the oxidation of SCFAs. The amount and relative abundance of SCFAs produced by carbohydrate fermentation are generally considered to be biomarkers of a healthy state [25]. Among SCFAs, butyrate plays a crucial role in maintaining gut epithelial integrity.

The gut epithelium serves as a barrier against pathogenic microorganisms and their products. The first line of this barrier is a two-layered mucus layer constructed by the goblet cell-secreted mucin 2 (MUC2) protein [26]. The inner mucus layer functions as a shield against the invasion of microbes and contains antimicrobial peptides and immunoglobulin A. The outer mucus layer is formed by the conversion of the inner mucus layer by the proteolytic processing of MUC2. Polysaccharides of MUC2 may be used as an energy source by some species of adherent microbes on the outer mucus layer. The expansion of mucin-degrading bacterial species may occur when dietary fiber is deficient in the gut. Although mucin degradation is a part of normal intestinal cell turnover [27], excessive degradation leads to the thinning of the mucus layer and gut epithelial barrier dysfunction [26]. Regarding colonic transit, mucin lubricates the alimentary tract and facilitates the passage of stools [15,28]. The second line of the barrier is epithelial cells sealed by multi-protein junctional complexes known as tight junctions (TJ). Butyrate promotes the production of mucin and TJ proteins and contributes to gut epithelial integrity [23,24,25]. Butyrate-producing bacteria in the colon are predominantly found in the families Lachnospiraceae and Ruminococcaceae [27]. Substrates that stimulate butyrate formation include starch, arabinoxylan-rich whole grains, and brans from cereals, such as wheat, rye, and oats [27].

### 3.2. Protein Fermentation and Uremic Toxins

Although limited information is currently available on the dietary proteins that escape digestion in the small intestine, 6–18 g of proteins is estimated to reach the colon daily [29]. Undigested peptides are broken down by proteolytic bacteria and subsequently used in protein fermentation or to form bacterial cell components. Culture-based experiments suggested that colonic bacteria preferentially ferment peptides over amino acids [30]. Protein fermentation produces hydrogen, carbon dioxide, methane, hydrogen sulfide, BCFAs such as valerate and caproate, ammonia, *N*-nitroso compounds, amines, phenols, and indoles [29,30]. Some of these metabolites are precursors of harmful uremic toxins.

Aronov et al. reported that several uremic toxins were almost absent in HD patients who underwent total colectomy, suggesting that an important proportion of uremic toxins is produced by bacterial proteolytic fermentation in the colon [31]. Indoxyl sulfate (IS), p-cresyl sulfate (PCS), and trimethylamine-*N*-oxide (TMAO) are well-known gut-derived uremic toxins (GDUTs), and their features are summarized in Table 1 [32,33,34,35,36]. IS is produced by tryptophan metabolism. Colonic bacteria, such as *Escherichia coli*, have tryptophanase, which converts tryptophan to indole. Since tryptophanase activity is enhanced at higher pH [37], the gut environment in CKD may facilitate the production of indole. After intestinal absorption, indole is sulfated to IS in the liver. PCS is produced from phenylalanine and tyrosine catabolism. The breakdown of these aromatic amino acids by anaerobic bacteria, such as Bacteroides, Enterobacter, and Clostridium, generates phenols and p-cresol [35]. p-Cresol is absorbed in the gut and metabolized to PCS in the liver. TMAO derives from the bacterial metabolism of quaternary amines, including betaine, l-carnitine, and phosphatidylcholine, which releases trimethylamine. Trimethylamine is absorbed in the circulation and oxidized to TMAO by flavin monooxygenase enzymes in the liver [36].

The renal clearance of IS and PCS depends on tubular secretion rather than glomerular filtration [33]. The protein-binding affinity of these toxins affects their removal from the circulation by conventional HD: TMAO is efficiently removed [32], whereas IS and PCS are not [33] (Table 1). These uremic toxins induce oxidative stress and pro-inflammatory responses [35,36], and their deleterious effects on CVD and CKD have been extensively reviewed elsewhere [36,38,39]. Although information on BCFAs is limited, increased levels of plasma valerate were independently associated with a history of CVD in a cross-sectional study including 214 CKD patients [40].

A healthy, balanced gut microbiota is primarily saccharolytic. The relative amount of saccharolytic and proteolytic fermentation is primarily regulated by the ratio of carbohydrates to nitrogen [41,42]. Since the digestion and absorption of proteins are impaired in CKD patients [43], a larger amount of proteins reaches the colon. Moreover, fiber-rich foods are restricted to avoid hyperkalemia in this population. Therefore, bacterial metabolism in CKD shifts to a proteolytic fermentation-dominant pattern. Colonic transit time is also associated with bacterial fermentation [44,45].

## 4. Intestinal Derangements in CKD

### 4.1. Gut Dysbiosis

Several CKD-related alterations in the intestinal tract have been suggested to cause gut dysbiosis [46]. The accumulation of urea in body fluids results in its massive influx into the intestinal tract. Urea is hydrolyzed to ammonia by bacterial urease. Ammonia is subsequently converted to ammonium hydroxide, which increases the intestinal luminal pH. Furthermore, significant amounts of uric acid and oxalate are secreted into the intestinal lumen in an adaptive response to the decline in their renal excretion. Urea and uric acid released in the intestines serve as alternative substrates for bacterial species that normally utilize indigestible carbohydrates. In addition, the dietary restriction of fiber-rich foods alters the bacterial composition and fermentation. Various medications used in CKD may also affect the gut microbiota.

The total quantity of fecal bacteria was previously reported to be lower in CKD/ESKD patients than in healthy controls [47,48,49]. In addition, the composition of the gut microbiota was significantly altered under these conditions. Vaziri et al. examined bacterial species in the stool samples of HD patients and age-, sex-, and ethnicity-matched control subjects using a phylogenetic microarray technique [50]. A significant difference was observed in the abundance of 190 bacterial operational taxonomic units between these groups, with the largest increases in the phyla Actinobacteria, Firmicutes, and Proteobacteria in HD patients. To isolate the influence of comorbidities, diet, and medication, the gut microbiota was investigated in CKD rats that underwent 5/6 nephrectomy, and the most notable finding obtained was decreases in the families Lactobacillaceae and Prevotellaceae [50]. In the secondary analysis of this study, Wong et al. identified a significant expansion of bacterial families possessing urease (such as Proteobacteria, Actinomycetales, and Clostridiaceae) and uricase (Proteobacteria and Actinomycetales) in HD patients [51]. In addition, bacterial families possessing indole- and p-cresol-forming enzymes (Clostridiaceae and Enterobacteriaceae) expanded, whereas those possessing butyrate-forming enzymes (Lactobacillaceae and Prevotellaceae) contracted. Jiang et al. also reported decreases in butyrate-producing bacteria, such as Roseburia, Faecalibacterium, Clostridium, Coprococcus, and Prevotella, in ESKD patients [47]. The abundance of the genera Lactobacillus and Bifidobacterium as well as *Klebsiella pneumoniae* was significantly lower in PD patients than in healthy controls [52]. Thus, the contraction of bacterial species producing lactate and butyrate and the expansion of those producing the precursors of GDUTs are considered to be representative findings of compositional alterations in CKD-related gut dysbiosis. However, conflicting findings have also been reported. The abundance of Lactobacillus [53] and Bifidobacterium [48] were unexpectedly greater in the fecal samples of CKD patients than in those of healthy controls in some studies. This inconsistency may be attributed to differences in ethnicity, diets (dietary restriction), and medications.

Regarding functional alterations in CKD-related gut dysbiosis, limited information is currently available on SCFAs. The relative abundance of butyrate-producing bacteria negatively correlated with an inflammatory marker in patients with CKD or ESKD [47,54]. Wang et al. reported that serum and fecal levels of SCFAs, particularly those of butyrate, were significantly lower in ESKD patients than in healthy controls [49]. In addition, fecal butyrate levels positively correlated with renal function. 

Although further studies are needed to characterize the compositional and functional alterations associated with CKD-related gut dysbiosis, the findings of an in vitro study by Van der Meulen et al. [55] are noteworthy. They showed that Bifidobacterium strains were capable of degrading aromatic amino acids in the absence of carbohydrates. Other experimental studies on *Prevotella copri* are also of value. *P. copri* was associated with better glucose tolerance in mice fed a fiber-rich diet [56], but with glucose intolerance in those fed a high-fat diet [57]. The dietary restriction of fiber-rich foods may be an important factor contributing to CKD-related gut dysbiosis and markedly affecting colonic bacterial functions.

### 4.2. Disruption of the Gut Epithelial Barrier

The disruption of the gut epithelial barrier is another intestinal derangement related to CKD. Shi et al. showed increased intestinal permeability and detected gut-derived bacterial DNA fragments in the blood of ESKD patients [58]. Experimental studies showed that urea [59], ammonia [59], and uric acid [60,61] impaired the gut epithelial barrier. The depletion of epithelium-protective mucin and gut epithelial TJ proteins was reported in CKD rats [62,63]. Various factors, such as pro-inflammatory cytokines [64], pathogenic bacteria [65], lipopolysaccharides (LPS) [66], nitric oxide [67], and butyrate production, regulate gut epithelial TJ proteins. These would be involved in the disruption of the gut epithelial barrier in CKD. Furthermore, gut epithelial edema and intestinal ischemia triggered by hypotension during and after HD sessions may further aggravate this disruption [46]. The translocation of microbial fragments, LPS, and luminal deleterious products occurs, and local and systemic inflammation is consequently induced. Local inflammation further amplifies the barrier disruption, thereby forming a vicious cycle.

Due to pro-inflammatory uremic toxins and the disruption of the gut epithelial barrier, gut dysbiosis is one of the main causes of chronic inflammation in CKD patients, which is closely related to CVD and strongly influences patient outcomes [68].

## 5. Pathogenic Factors Linking Gut Dysbiosis to Constipation in CKD

In the 1980s, chronic inflammation throughout the alimentary tract of HD patients was reported in an autopsy study [69]. Intestinal inflammation is known to play an important role in intestinal dysmotility [70,71]. In a recent experimental study by Yu et al., intestinal motility was significantly decreased in CKD rats that underwent 5/6 nephrectomy compared to control rats [72]. Furthermore, the expression levels of interleukin-6, tumor necrosis factor-α, and inducible nitric oxide synthesis (iNOS) were increased in the gut of CKD rats. Although data were not shown, the authors also reported that an iNOS inhibitor restored intestinal motility [72]. In addition, Nishiyama et al. revealed the involvement of gut dysbiosis in inflammation and dysmotility of the gut [73]. In their study, CKD mice showed gut dysbiosis, intestinal dysmotility, a reduced fecal amount, and intestinal inflammation. Notably, antibiotic therapy improved intestinal dysmotility and increased the fecal amount and the expression of inflammatory cytokines.

In this section, we discuss the mechanisms linking CKD-related gut dysbiosis to constipation with a focus on inflammation and bacterial fermentation products.

### 5.1. GDUTs

Hoibian et al. showed that the gastrointestinal transit time of adenine-induced CKD mice was 1.8-fold longer than that of control mice [74]. Resected colons from control mice were incubated with the plasma of HD patients or healthy subjects. Colons incubated with uremic plasma exhibited a lower level of maximal contraction than those incubated with healthy plasma. Colons were subsequently incubated with uremic toxins at the concentration encountered in ESKD patients (urea at 50 mmol/L, IS at 200 μmol/L, or PCS at 200 μmol/L) for 2 h. The incubation with IS or PCS, but not with urea, significantly decreased the maximal force of colonic contractions. Thus, accumulated GDUTs may play an important role in constipation/intestinal dysmotility. The findings of clinical studies support this hypothesis. In 43 non-dialysis CKD patients, Ramos et al. reported that increased serum PCS levels were associated with constipation defined by BSFS type 1–2 stools [75]. Similar findings were reported in patients treated with HD [76] and PD [77]. These toxins may induce intestinal dysmotility via their pro-inflammatory effects.

Homocysteine, an intermediate in methionine metabolism, is also regarded as a GDUT [78]. In adenine-induced CKD rats, the administration of homocysteine impaired the gut epithelial barrier and increased intestinal permeability by stimulating inflammatory and oxidative damage [79]. However, intestinal motility was not evaluated in this study. Notably, animal models with hyperhomocysteinemia induced by the heterozygous knockout of cystathionine β-synthase showed decreased intestinal motility [80]. In addition, we reported that the increased plasma level of total homocysteine in HD patients was associated with constipation after adjustments for age, sex, and DM [81]. Homocysteine is remethylated to methionine via the remethylation pathway or irreversibly degraded via the transsulfuration pathway [82]. These pathways depend on vitamins B6, B9 (folate), and B12, which are produced by the genera Bifidobacterium and Lactobacillus [83].

Thus, GDUTs may induce intestinal dysmotility in an inflammation-dependent manner. On the other hand, some uremic toxins may induce gut dysmotility independently of inflammation. Nishiyama et al. incubated resected colons from normal mice with spermine and observed decreased intestinal contractions [73]. Spermine is a uremic toxin classified to protein-bound polyamines [84]. Polyamines function as agonists for calcium-sensing receptors in Auerbach’s plexus and inhibit intestinal motor activity [85]. However, the concentration of spermine used in the incubation in Nishiyama’s study was markedly higher than plasma spermine levels reported in CKD patients [86]. Spermine does not affect intestinal motility in clinical practice. On the other hand, some GDUTs may induce intestinal dysmotility in an inflammation-independent manner. This issue warrants further study.

The pathogenic relationship between GDUTs and constipation needs to be elucidated in more detail. The findings of in vitro and human studies suggested that a long colonic transit time is associated with a shift in bacterial metabolism from saccharolytic to proteolytic fermentation [44,45]. Decreased intestinal motility may affect uremic toxin generation by increasing the availability of amino acids to be fermented in the colon. Accumulated uremic toxins, in turn, exacerbate intestinal dysmotility, thereby forming a vicious cycle.

### 5.2. Butyrate-Related Mechanisms

Butyrate exerts anti-inflammatory effects and plays a crucial role in maintaining the gut epithelial barrier by increasing the production of mucin and TJ proteins [23,24,25]. Reduced intestinal inflammation and enhanced lubrication in the alimentary tract lead to rapid colonic transit. Moreover, butyrate has been shown to promote intestinal motility [15,23,25]. Soret et al. demonstrated that butyrate stimulated in vivo colonic transit and ex vivo cholinergic-mediated contractile responses [87]. As described above, butyrate-producing colonic bacteria [47,51] and the serum and fecal levels of butyrate [49] were significantly decreased in CKD patients. Decreased butyrate production may contribute to CKD-related constipation.

## 6. Potential Interventions for Gut Dysbiosis-Related Constipation

The expected goals of the following interventions include regular bowel movement, improvements in stool consistency and incomplete evacuation, the amelioration of the gut microbial composition, and a decrease in the synthesis of harmful uremic toxins.

### 6.1. Probiotics, Prebiotics, and Synbiotics

Probiotics are live microorganisms that confer health benefits to the host. Lactobacillus and Bifidobacterium, which lower colonic luminal pH through the production of acetate and lactate and inhibit the growth of pathogens [88], are frequently adopted as constituents of probiotics. Prebiotics are indigestible food ingredients that include dietary fiber, oligosaccharides, polysaccharides, and resistant starches. Synbiotics refers to a combination of probiotics and prebiotics. A meta-analysis by Dimidi et al. revealed that probiotics reduced the whole-gut transit time in non-CKD adults with functional constipation by 12.4 h and increased stool frequency by 1.3 times/week [89]. Stool consistency also improved. Similarly, another meta-analysis showed that prebiotics and synbiotics increased stool frequency and improved its consistency [90]. Previous studies on CKD reported the beneficial effects of prebiotics and synbiotics on bowel habits, but only included a small number of subjects [76,91,92]. Although dietary fiber is a main source of prebiotics, its mean intake previously reported in HD patients ranged between 5.0 and 16.6 g/day [2]. Soluble fiber accelerates colonic transit via hydrophilic properties and the osmotic effects of fermentation byproducts, and insoluble fiber by increasing the stool biomass, leading to the direct stimulation of intestinal secretion and motility [93]. However, an increased fiber intake may be insufficient to improve bowel habits in a large proportion of CKD patients. Voderholzer et al. reported that fiber supplementation attenuated the symptoms of constipation in 85% of patients with normal transit constipation, whereas 63% of those with slow transit constipation did not respond to this therapy [94]. In addition, some patients may experience abdominal distention or flatulence induced by fiber supplementation in a dose-dependent manner [93]. The effects of these agents on bowel movement in CKD patients need to be examined in large-scale trials.

In a meta-analysis of 16 randomized controlled trials (RCTs) on CKD patients (probiotics in 6, prebiotics in 5, and synbiotics in 5), McFarlane concluded that these agents did not decrease serum levels of urea, IS, or PCS, except for a slight decrease in urea levels by prebiotics [95]. In two studies investigating changes in bacterial compositions, synbiotics significantly increased the abundance of Bifidobacterium [95]. Other authorities also mentioned that that probiotics were inefficient to modulate uremic toxins or systemic inflammation [46,96]. In order to obtain the benefits of probiotics, the administration of symbiotic bacteria in combination with improvements in the gut biochemical milieu appears to be necessary [46]. A carbohydrate deficiency may hinder probiotics from exerting their beneficial effects, as shown by Van der Meulen et al. [55]. Therefore, Mafra et al. recommended prebiotics and synbiotics rather than probiotics to reduce inflammation and GDUT production [20]. Prebiotics are found naturally in many fruits, cereals, and vegetables (such as asparagus, sugar beet, garlic, chicory, onion, and banana), but are restricted in CKD patients. The bacterial fermentation of prebiotics stimulates the growth of Bifidobacterium and Lactobacillus at the cost of other bacteria, such as Bacteroides and Clostridium [20]. The beneficial effects of synbiotics depend on a combination of the probiotics and prebiotics selected. Further studies are needed to establish a more effective and sophisticated prescription of these agents.

### 6.2. Dietary Intervention

Multiple dietary restrictions have been recommended for CKD patients, particularly for those undergoing dialysis therapy. However, these restrictions do not meet cardioprotective dietary recommendations [97,98]. The restriction of plant-based foods reduces the intake of beneficial vitamins, magnesium, and dietary fiber and potentially worsens hypertension, hyperphosphatemia, and metabolic acidosis [99]. In recent observational studies, vegetable-rich diets were associated with a lower risk of incident CKD [100,101,102]. In dialysis patients, a low fiber intake was associated with inflammation, oxidative stress, arterial stiffness [103], cardiovascular events [104], and higher mortality [105]. The need for a paradigm shift in dietary therapy for CKD is being increasingly discussed [20,99,106,107,108].

Appropriate cooking procedures, such as soaking and boiling, need to be employed in order to increase fiber intake and decrease the potassium load [108,109]. Potassium removal by home-based cooking methods has been summarized in a review by Cupisti et al. [109]. Additionally, salt substitutes and food additives need to be considered [109,110]. In some cases, 5–6 g of sodium substitutes contains 1000–1800 mg of extra potassium [109]. Potassium-based additives are being increasingly used in the food supply as preservatives, flavor enhancers, antioxidants, stabilizers, and emulsifiers [110]. The potassium content of foods with additives varies widely, with 100 g of a smoked ham product containing 1500 mg of potassium [111]. The bioavailability of potassium from whole vegetables and fruits is low at 50–60%, whereas that from food additives may reach 100% [110], and, thus, serves as a hidden cause of hyperkalemia.

### 6.3. Formulations of SCFAs

Due to their beneficial effects on the host, the therapeutic uses of SCFAs are attracting increasing attention. The effects of the administration of butyrate have been investigated in inflammatory bowel disease [23]. Wang et al. showed that the administration of butyrate reduced renal histopathological injuries and serum TMAO levels in CKD rats [49]. In an experimental study by Gonzalez et al., sodium butyrate increased the production of mucin, gut epithelial TJ proteins, and anti-inflammatory interleukin-10, decreased circulating LPS, and improved renal function and insulin resistance [63]. In a clinical study by Marzocco et al., sodium propionate improved insulin resistance, inflammation, and oxidative stress and reduced plasma levels of IS and PCS in 20 HD patients [112]. On the other hand, the clinical effects of butyrate have not yet been examined in CKD. Its unpleasant taste and odor make the oral administration of butyrate formulations difficult [23]. The safety of SCFA administration is another issue that needs to be resolved. Tirosh et al. suggested that propionate may induce insulin resistance [113], which is inconsistent with the findings reported by Marzocco [112]. Further studies are required to understand both the benefits and harmful effects of SCFAs.

### 6.4. Fecal Microbiota Transplantation

Tian et al. recently reported that fecal microbiota transplantation (FMT) improved the colonic transit time as well as stool frequency and consistency in a RCT on 60 patients with slow transit constipation [114]. The effective colonization of donor microbiota was observed in recipients one year after FMT [115]. These effects of FMT indicate a close relationship between gut dysbiosis and bowel habits. However, it is unlikely to be employed for the treatment of constipation in clinical practice because of the cost and complexity of the procedure.

### 6.5. Laxatives

Although CKD patients with constipation are predominantly treated with stimulant laxatives [81,116], the American Gastroenterological Association recommends the temporal use of these agents [117]. Intestinal secretagogues, such as lubiprostone, linaclotide, elobixibat, and tenapanor, appear to be more effective for the management of bowel habits in this population. These agents are attracting attention due to their pleiotropic effects.

Lubiprostone activates chloride channels on the epithelium of the small intestine and increases water secretion into the intestinal lumen, resulting in accelerated intestinal transit. In HD patients, lubiprostone significantly increased stool frequency from 1.8 ± 1.3 to 4.5 ± 1.5 times/week [118]. This agent also increased gastrointestinal mucin secretion [119,120] and exerted protective effects on the gut epithelial barrier [121,122]. In adenine-induced CKD mice, lubiprostone increased the abundance of the family Lactobacillaceae and genus Prevotella, decreased plasma IS levels, and attenuated renal fibrosis and inflammation [123]. In a recent experimental study, linaclotide, a guanylate cyclase C agonist, ameliorated the expression of the gut epithelial TJ protein, plasma TMAO levels, and macrophage infiltration in the small intestine, with reductions in cardiac and renal fibrosis [124]. The elevated abundance of bacteria, such as Clostridiales and Corynebacterium, was decreased by linaclotide. On the other hand, Sueyoshi et al. showed a prebiotic effect of lactulose [125], a disaccharide of galactose and fructose. They demonstrated that lactulose improved oxidative stress, serum IS levels, and renal injuries in adenine-induced CKD rats. Additionally, it decreased the abundance of indole-producing bacteria, such as Bacteroides. Further studies are needed to clarify whether these laxatives clinically exert these pleiotropic effects.

Gen et al. reported that after the administration of lubiprostone for 3 months, serum phosphate levels significantly decreased from 4.7 ± 1.5 (1.52 ± 0.48) to 3.8 ± 1.1 mg/dL (1.23 ± 0.36 mmol/L) [118]. A similar effect of elobixibat, an inhibitor of the ileal bile acid transporter, was reported by Shono et al. [126]. Accelerated intestinal transit may inhibit the intestinal absorption of phosphate [118] and potassium [127], which is favorable for reducing cardiovascular complications in CKD patients. Tenapanor, an inhibitor of sodium/hydrogen exchanger 3, may be preferred by clinicians because it inhibits the intestinal absorption of phosphate and sodium [128]. In CKD rats, tenapanor therapy decreased serum levels of creatinine, phosphate, and fibroblast growth factor-23 and attenuated vascular calcification and cardiac hypertrophy [129].

### 6.6. Enhancements in the Dialytic Removal of GDUTs

Due to their high protein-binding properties, IS and PCS cannot be adequately removed by conventional HD (Table 1). Ibuprofen shares the same primary albumin-binding site with these toxins. The drug dissociates these toxins from albumin molecules and increases their free fraction. Madero et al. infused ibuprofen into the arterial bloodline during a HD session in 18 patients and observed a significant increase in the dialytic removal of IS and PCS [130]. In an experimental study using uremic rats, a lipid emulsion formulation similarly increased the dialytic removal of these toxins [131]. In an in vitro study by Yamamoto et al., direct hemoperfusion using a column containing activated carbon effectively reduced the concentrations of these toxins in the circulation [132]. However, due to the intermittent nature of HD therapy, these methods may need to be combined with therapeutic options that inhibit the production of these uremic toxins.

### 6.7. Oral Adsorbents

AST-120 is an oral charcoal adsorbent that consists of black spherical particles of 0.2–0.4 mm in diameter. Indole, p-cresol [133], and ammonia [134] were previously reported to be adsorbed by this drug. AST-120 partially restored gut epithelial TJ proteins and reduced plasma endotoxin levels with the attenuation of inflammation and oxidative stress in CKD rats [135], but failed to prevent the progression of CKD in a large multinational RCT [136]. In addition, the adherence to this drug is generally poor due to its gastrointestinal adverse effects, including constipation.

Sevelamer, a non-absorbable anion exchange resin, is used as a phosphate binder in CKD patients. It adsorbs substances other than phosphate, which may explain the mechanism underlying its various pleiotropic effects [137], but does not appear to effectively adsorb IS or PCS [137,138]. Constipation is a well-known adverse effect of sevelamer.

### 6.8. Exercise Therapy

Although we have a limited understanding of the effect of exercise on the gut microbiota, Ortiz-Alvarez et al. showed that higher levels of physical activity and cardiorespiratory fitness were positively associated with bacterial diversity, the counts for Bifidobacterium and Prevotella, and SCFA concentration in subjects’ stools [139]. In a meta-analysis, Gao et al. suggested that exercise may be a feasible and effective strategy to ameliorate constipation-related symptoms and quality of life in the non-CKD population [140]. To the best of our knowledge, it currently remains unknown whether exercise has a similar effect in CKD patients. However, exercise therapy was recently suggested to improve physical function [141,142] and restless legs syndrome [143] in dialysis patients. Even if the effect of exercise on constipation is not sufficient, habitual exercise is associated with other beneficial effects.

## 7. Conclusions

Gut dysbiosis was recently suggested to be associated with various CKD-related disorders, such as insulin resistance [144], malnutrition [145], sarcopenia [144], frailty [54], bone disorders [146], and chronic inflammation. As reported by recent studies, gut dysbiosis appears to be closely related to the pathogenesis of constipation in CKD. CKD-related constipation is often intractable due to its multifactorial nature. Every available means needs to be employed to treat constipation. Gut dysbiosis and increased levels of plasma GDUTs emerge in relatively early stages of CKD [147]. As CKD progresses, gut dysbiosis, uremic toxin accumulation, and intestinal dysmotility may worsen in a vicious cycle. Further studies are needed to elucidate the pathogenic relationship between gut dysbiosis and constipation in CKD, which may lead to novel treatment options not only for constipation, but also for chronic inflammation, CVD, and CKD itself.

## Figures and Tables

**Figure 1 microorganisms-08-01862-f001:**
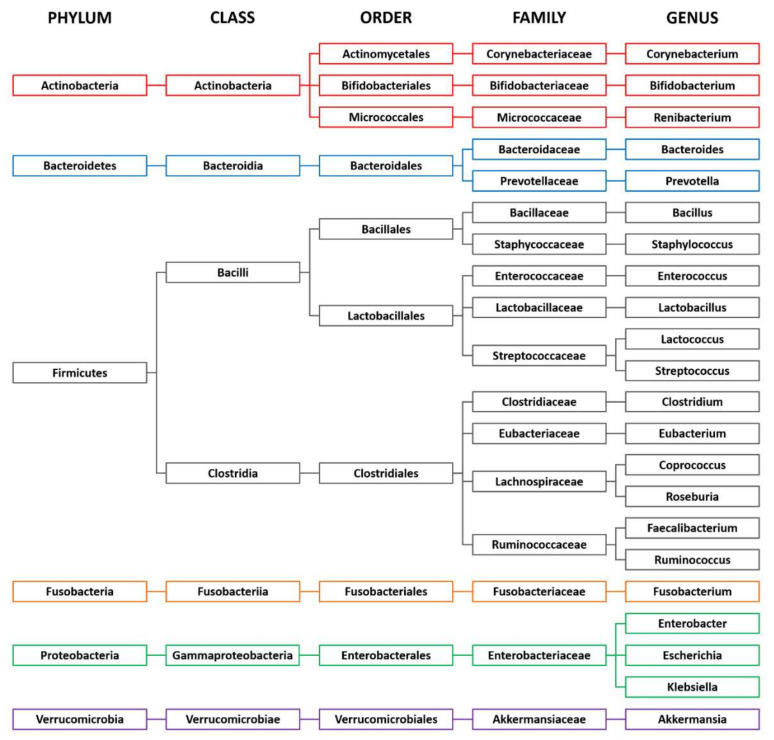
Examples of bacterial taxonomy in the gut.

**Table 1 microorganisms-08-01862-t001:** Features of gut-derived uremic toxins.

Uremic Toxin	Substrates (Main Food Materials)	MW (Da)	Protein-Bound	Mean Reduction * by HD (%)
IS	Tryptophan (egg white, meat, milk, cheese, soy product)	213	Yes	36
PCS	Tyrosine (egg white, meat, milk, cheese, soy product) Phenylalanine (egg, meat, fish, milk, cheese, nut, legume)	188	Yes	31
TMAO	L-carnitine (red meat) Choline (egg yolk)	75	No	59

Abbreviations: HD, hemodialysis; IS, indoxyl sulfate; MW, molecular weight; PCS, p-cresyl sulfate; TMAO, trimethylamine-*N*-oxide. * Reduction rate (%) = (pre-concentration – post-concentration) / pre-concentration × 100.

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
