# Peer review of "Chronic Kidney Disease, Gut Dysbiosis, and Constipation: A Burdensome Triplet"

_microorganisms, 2020, doi:10.3390/microorganisms8121862_

Round 1
Reviewer 1 Report
Thank you so much for offering me an opportunity to review the manuscript entitled “Chronic Kidney Disease, Gut Dysbiosis, and Constipation: A Burdensome”. This paper demonstrated and summarized the importance of gut dysbiosis as a therapeutic target in CKD patients. The authors also showed the therapeutic options against CKD-related dysbiosis. This paper is of clinically very helpful for better understanding how important gut microbiota and dysbiosis are in progression of CKD. This paper looks well-organized, but there is a small issue to be resolved.
Major concern,
- In figure2, authors showed that gut epithelial tight junction proteins is reduced only by dysbiosis-induced decrease in butylate. However, not only butylate production, but also other factors such as proinflammatory cytokines, pathogenic bacteria, LPS, and nitric oxide are regulating tight junction proteins in intestinal tract [Ma et al. 2004][Lemichez E et al. 2010][Rocha et al. 2019]. Is only the decrease in butylate production explainable for TJ proteins dysregulation in gut epithelium? Please explain or add some more information about the causes of the decreased TJ proteins in CKD-related dysbiosis in Figure 2.
Reviewer 2 Report
Chronic Kidney Disease, Gut Dysbiosis, and Constipation: A Burdensome Triplet is a novel and very interesting topic. But the review still has to be revised per items as follows:
- CKD is the core of this review, so the related characterizes of CKD, like symptoms and underlying mechanism should be summarized in the introduction.
- Figure 1, being changed to a table will be simpler and clear with less space.
- In the introduction, the author mentioned physical activity decrease could be one of the multiple factors of CKD-related constipation. Why did not discuss the effects of lifestyle changes and consider it as one of potential prevention?
- Language checking is needed.
